# Prediction Model for the Evolution of Residual Stresses and Machining Deformation of Uneven Milling Plate Blanks

**DOI:** 10.3390/ma16186113

**Published:** 2023-09-07

**Authors:** Yaohui Zheng, Pengcheng Hu, Minghai Wang, Xiaoyue Huang

**Affiliations:** Sohool of Mechatronics Engineering, Shenyang Aerospace University, Shenyang 110136, China; zhengyh214@sau.edu.cn (Y.Z.); wangminghai2008@163.com (M.W.); huangxiaoyue@stu.sau.edu.cn (X.H.)

**Keywords:** analytical model, initial residual stress, uneven milling, machining deformation

## Abstract

During aerospace thin-walled component processing, the prediction and control of machining deformation have gained increasing attention. The initial residual stress in the blank is a major factor leading to the occurrence of machining deformation. This paper proposes the concept of uneven milling during the workpiece machining process, which is caused by the variation in local cutting depth resulting in uneven material removal thickness. Based on the elasticity theory, an analytical model is established to predict the evolution of overall residual stress and machining deformation in beam-like aluminum alloy components under uneven milling conditions. The effectiveness of the model is verified through finite element simulations and experiments. The results are as follows: (1) Under uneven milling conditions, the analytical model can accurately predict the distribution of residual stress and the machining deformation within the ZX section of the workpiece. (2) The uneven distribution of bending stress arises from the different curvature radii of various positions after workpiece deformation, leading to a 1 MPa to 3 MPa difference in stress between the middle and both ends of the workpiece. (3) During the layer-by-layer milling process, the magnitude of workpiece deformation is related to the stress state of the material removed, and there is a deformation superposition effect on the lower surface of the workpiece, further exacerbating the overall machining deformation.

## 1. Introduction

Widespread application of aluminum alloys has emerged in the design and fabrication of integrated thin-walled aerospace components, owing to their notable attributes of high strength, lightweight properties, and cost effectiveness [1]. However, the deformation issue during the machining of aerospace aluminum alloy structures has long been a formidable challenge in the aviation industry. Annually, dimensional deviations caused by material removal rates reach as high as 90% and the characteristic thin-walled design of aluminum alloy integrated components results in billions of dollars lost for the aviation manufacturing sector [2]. During the progressive removal of material layers, the internal stress state of the billet changes. Initially, as material layers are removed, the equilibrium state is disrupted, leading to stress redistribution aimed at restoring equilibrium. This process, in turn, induces billet deformation. Research indicates that factors such as initial residual stress (IRS), machining-induced residual stress(MIRS), clamping force, and significantly cutting the load influence the redistribution of internal stress within the billet during material removal, and, thus, can be considered the primary causes of workpiece deformation [3]. Existing studies have focused on these factors as starting points, proposing predictive models and control strategies for machining deformation.

Scholars have extensively discussed the mechanisms of inducing deformation through IRS and proposed analytical models and deformation control methods for predicting such deformation. Virkar et al. [4], based on the theory of beam bending, proposed a theoretical model for back calculating the IRS in a workpiece, which effectively handled asymmetric stress distribution situations. Shin [5] presented a theoretical model for predicting the deformation caused by uniaxial IRS in the process of aluminum alloy layer removal. This model considers the material’s elastic properties, provides a quantitative description by analyzing the influence of IRS on the deformation during the process, and offers a theoretical framework for numerous researchers. Gao et al. [6], based on plate–shell theory, introduced a predictive model for plate deformation that considers biaxial IRS. Moreover, they verified that symmetrical processing helps suppress deformation and discovered that the position of the workpiece within the blank significantly affects deformation.

In recent years, research focused on the coupled influence of MIRS and IRS on machining deformation has increased. By combining numerical calculations of equivalent stiffness with plate shell theory, Li et al. [7] proposed a novel predictive model for machining deformation that considers the effects of both MIRS and IRS. This model quantitatively expresses the impact weights of IRS and MIRS on maximum machining deformation, resulting in an approximate 10% accuracy improvement compared to deformation predictive models that solely consider IRS [8]. It is worth noting that, based on experimental analysis conducted on six typical thin-walled components, the contribution of IRS to machining deformation exceeded 85%. Wang et al. [9] established an analytical model for predicting strain energy to anticipate machining deformation in lattice components, which incorporates the influence of IRS and MIRS. The authors also proposed a method to mitigate machining deformation through adjustments to the machining reference. Additionally, they suggested selecting a machining position within the billet material with the lowest strain energy prior to rough machining to minimize the impact of IRS on machining deformation. Based on the principles of material mechanics and response surface design methodology, Jiang et al. [10] incorporated milling parameters into a deflection calculation model, enabling the prediction of residual stress distribution in workpieces under different cutting parameters and deriving deformation deflection caused by machining.

Much research has revealed the influence of initial residual and MIRS on machining deformation through simulation and experimental methods. Huang et al. [11] identified that under the coupling effect of initial compressive stress and MIRS, sheet metal deformation increases, while under the coupling effect of initial tensile stress and MIRS, this deformation decreases. Yang et al. [12] proposed that in the elastic region, the IRS adds linearly to the MIRS along the depth. However, in the plastic region, MIRS is scarcely affected by IRS. Madariaga et al. [13] established a simplified finite element simulation model for component deformation machining considering IRS and MIRS, based on IRS measured using the contour method and MIRS measured using the blind-hole method. The authors also verified the significant influence of different machining strategies on deformation. Chighizola et al. [14] used aluminum alloy samples with different IRSes and combined simulation with material strength models to validate that the milling deformation maximum curvature direction depends on the sign and direction of the IRS. Cerutti et al. [15,16] developed a finite element tool based on Boolean operations for simulating the material removal processes of complex structural components. Compared to the element birth-and-death method, this option more accurately predicts residual stress distribution inside the workpiece and deformation during the machining process.

Theoretical research provides the foundation and guidance for controlling machining deformation. Fan and Yang [17,18] proposed an optimization method based on the energy release mechanism during the machining process to improve the material removal sequence. This method suggests removing the high-strain energy material elements inside the workpiece during the rough machining stage and correcting the machining reference before the finish machining stage to effectively control deformation during subsequent machining. The effectiveness of this method was verified through milling experiments on beam-type and plate-type workpieces. Using experiments, Yang et al. [19] identified that during the aluminum alloy frame part milling process, deformation induced by the web accounted for over 74% of the total deformation, while deformation induced by the sidewall accounted for less than 26%. Zhang et al. [20] optimized the workpiece positional distribution in the raw material based on the residual stress contour map obtained from the contour method, aiming to minimize the influence of machining deformation.

Moreover, the overall residual stress inference within the blank during processing is of significant importance for deformation control. Wang et al. [21] obtained the blank overall residual stress field by incorporating surface stress measurement results into finite element simulation units and the overall residual stress equilibrium equation. This method avoids material loss caused by destructive testing and was validated using beam-like component verification. Fu et al. [22] proposed a segmental calculation method for IRS based on superposition theory and the improved crack compliance method, which was verified for accuracy through simulation and experimentation. Wang et al. [23] combined deformation force monitoring data during the machining process with finite element models and introduced the strategy gradient algorithm to iteratively calculate the inferred IRS field. Jovani et al. [24] proposed a method for measuring deformation caused by stress release layer by layer during the removal process using noncontact optical measurement, thus, deducing the distribution of IRS. This method enables deformation measurement and workpiece stress prediction under clamping conditions. However, further research is still required for deformation measurement in complex structural components.

In previous studies, the exploration of shape deviations caused by local cutting depth variations during workpiece machining is limited. Lin et al. [25] established a mechanical model and utilized finite element analysis to investigate the influence of initial geometric errors of the blank on part deformation. They optimized the machining positions using a stepwise reduction algorithm. The research findings verified the significance of initial geometric errors of the blank on part deformation. Aurrekoetxea et al. [26] proposed a method for characterizing residual stresses during machining on machine tools. This method considers the coupled characteristics of ribbed geometries and enables two-dimensional representation of residual stresses while considering the stress induced during machining and initial deformation. Gulpak et al. [27,28] proposed a model for workpiece shape deviations caused by surface residual stress and thermal elastic effects during thin-walled component machining. This model suggests that local cutting depths vary owing to thermal expansion effects during the machining process. This variation can be explained by thermal accumulation and vertical displacement potential during machining. However, a thorough investigation of the machining deformation and stress distribution caused by the variation in material removal thickness during the machining process has not been conducted.

Therefore, this study introduces the concept of uneven milling, which refers to the uneven removal of material thickness caused by variations in local cutting depths. Based on the principles of elasticity, this study deduces the evolution of overall residual stress in beam-like components and a model for predicting machining deformations under the conditions of uneven milling is proposed. Finally, the model is validated using finite element simulations and machining experiments and the overall residual stress evolution patterns in workpieces under uneven milling are summarized.

## 2. Analytical Model for Prediction of Machining Deformation

This study presents an analytical model for machining deformation prediction, and puts forward the following four hypotheses:The billet is isotropic and homogeneous in nature.The influence of MIRS on machining deformation is ignored.IRS distribution at the same depth is uniform.Shear stresses influence is disregarded for the aluminum alloy beam undergoing pure bending deformation.

### 2.1. Model for Predicting Machining Deformation under the Uniform Milling Conditions

During the aluminum alloy billet forming process, residual stress primarily originates from forging, heat treatment, and cold working processes. These manufacturing processes induce billet stress and strain, which subsequently results in residual stress during cooling and as a result of uneven temperature distribution. Previous research has demonstrated that the deformation of beam-shaped aluminum alloy components primarily manifests as bending deformation along the longitudinal X-direction [19,29,30]. Therefore, this study primarily focuses on the influence of longitudinal (X-direction) IRS on the characteristics of deformation during processing and stress redistribution (this is expressed as $\sigma_{y}=\sigma_{z}=0$, where $\sigma_{y}$ and $\sigma_{z}$ represent the residual stress in the width Y direction and the thickness Z direction, respectively). As shown in Figure 1a, the raw material dimensions in the lengthwise X-direction are l mm, in the widthwise Y-direction are b mm, and in the thicknesswise Z-direction are d mm. Prior to material removal, the IRS in the raw material is in a state of equilibrium, satisfying the following equilibrium equations for forces and moments.
(1)$$F_{x}=b\int_{0}^{d}\sigma_{x}(z)dz=0$$
(2)$$M_{x}=b\int_{0}^{d}z\cdot \sigma_{x}(z)dz=0$$

Here, $F_{x}$ and $M_{x}$ represent the forces and moments in the X-direction, respectively, while $\sigma_{x}(z)$ represents the X-direction residual stress that varies with the thickness Z.

After removing a layer of thickness e from the raw material, the original state of equilibrium is disrupted. To ensure internal force balance is maintained, a uniform stress σ(e) will be generated. This stress can be regarded as a uniform stress distribution exerted by the raw material removed portion on the remaining portion. The uniform stress σ(e) is uniformly distributed along the cross-section of the remaining raw material and is expressed by Equation (3) as follows.
(3)$$\sigma (e)=\frac{b\int_{d-e}^{d}\sigma_{x}(z)dz}{b[d-e]}=\frac{-1}{d-e}\int_{0}^{d-e}\sigma (z)dz$$

The stress distribution $\sigma_{s}(z)$ [4] for the modified remaining portion of the raw material can be derived from the superposition principle. Equation (4) details the expression for $\sigma_{s}(z)$.
(4)$$\sigma_{s}(z)=\sigma_{x}(z)+\sigma (e)$$

As shown in Figure 1b, the modified stress distribution $\sigma_{s}(z)$ satisfies force balance albeit not moment balance, as shown in Equations (5) and (6).
(5)$$b\int_{0}^{d-e}\sigma_{s}(z)dz=0$$
(6)$$M_{x}(e)=b\int_{0}^{d-e}\sigma_{s}(z)zdz\neq 0$$

The moment $M_{x}(e)$ will cause the raw material to undergo bending deformation to satisfy the moment balance condition. This also results in the formation of a stress distribution $\sigma_{m}(z)$ inside the raw material that varies linearly with thickness Z, as shown in Equation (7).
(7)$$\sigma_{m}(z)=\frac{M_{x}(e)}{I}(\frac{d-e}{2}-z)$$

While I denotes the remaining unprocessed sections moment of inertia.
(8)$$I=\frac{b\cdot {\left(d-e\right)}^{3}}{12}$$

As shown in Figure 1c, the distribution of residual stress, denoted as $\sigma_{new}(z)$, can be expressed to fulfill both force and moment equilibrium conditions as follows.
(9)$$\sigma_{new}(z)=\sigma_{s}(z)+\sigma_{m}(z)$$

By substituting Equations (3), (4), and (6)–(8) into Equation (9) and solving it, it is possible to obtain the distribution function for the overall stress in the workpiece under the uniform thickness milling condition. Based on the assumption of pure bending deformation according to beam bending theory, the effect of shear stress on the buckling deformation is neglected. Therefore, the bending deformation caused by the workpiece internal stress imbalance can be equivalent to the deformation induced by a uniform torque applied at both ends of the workpiece, denoted as $M_{x}(e)$ [20]. The deflection curve can be obtained using the method of integration as follows:(10)$$\omega (x)=\iint [\frac{M_{x}(e)}{EI}dx]dx+C_{1}x+C_{2}$$

In Equation (10), EI represents the beam flexural rigidity, where E is the elastic modulus. By substituting the boundary conditions ω(0)=0 and ω(l)=0 into Equation (10), the undetermined coefficients C_1_ and C_2_ can be solved.

### 2.2. Model for Predicting Machining Deformation under the Uneven Milling Conditions 

During the machining process, the cutting depth, denoted as ‘e’, is typically maintained at a constant value. However, after removal of the first workpiece layer, the distortion caused by redistribution of residual stress can result in localized variations in the cutting depth during the second layer removal. Such variations further result in an uneven removal of material thickness, known as the uneven milling phenomenon. Section 2.1 derives the computational formulae for stress distribution and deflection curves after the first layer removal, and utilizes the calculated results as initial conditions for the removal of subsequent layers, thus deriving the expressions for stress distribution and deflection curves under uneven milling conditions. Hence, during the second layer removal, the initial workpiece deformation, denoted as ω(x), can be represented by Equation (9), while the initial stress distribution is given by ${\sigma^{\prime }}_{x}(z)=\sigma_{new}(z)$, as shown in Equation (8). This model utilizes the discretization method for cross-sections (considering cross-section *i* in Figure 2 as an example) and combines it with the equal-thickness milling model to derive the expressions for stress distribution and deflection curves under uneven milling conditions. In the range of 0 to l in the workpiece longitudinal direction, n cross-sections are selected (the normal direction of the section is the X-axis), and the stress distribution and deformation deflection results of the remaining workpiece part are calculated in each cross-section. By fitting the results of each cross-section calculation, the final stress surface and deflection curve can be obtained. Figure 2 illustrates the workpiece uneven milling mechanism from a Y-oriented perspective. In cross-section *i*, the initial deflection is denoted as $\omega (x_{i})$, and the initial stress distribution is represented by ${\sigma^{\prime }}_{x_{i}}(z)$.

According to Figure 2, before the second layer removal, the workpiece is in an equilibrium state. However, when the shaded portion is removed, the original equilibrium state is disrupted, leading to the redistribution of residual stress in the workpiece. Equation (11) is used to describe the resultant force acting on the remaining workpiece portion in cross-section *i* after material removal. This resultant force generates a uniform stress $\sigma_{x_{i}}(e)$ in cross-section *i*, as shown in Equation (12).
(11)$$b\int_{0}^{d-e-\omega (x_{i})}{\sigma^{\prime }}_{x_{i}}(z)dz=-b\int_{d-e-\omega (x_{i})}^{d}{\sigma^{\prime }}_{x_{i}}(z)dz$$
(12)$$\sigma_{x_{i}}(e)=\frac{-1}{d-e-\omega (x_{i})}\int_{0}^{d-e-\omega (x_{i})}{\sigma^{\prime }}_{x_{i}}(z)dz$$

According to the superposition principle, the modified stress distribution $\sigma_{s_{i}}(z)$ is given by Equation (13). However, it is important to note that this stress distribution satisfies the force equilibrium albeit not the moment equilibrium, as expressed in Equations (14) and (15).
(13)$$\sigma_{s_{i}}(z)=\sigma_{x_{i}}(z)+\sigma_{x_{i}}(e)$$
(14)$$b\int_{0}^{d-e-\omega (x_{i})}\sigma_{s_{i}}(z)dz=0$$
(15)$$M_{x_{i}}(e)=b\int_{0}^{d-e-\omega (x_{i})}\sigma_{s_{i}}(z)\cdot (z-z_{0}(x_{i}))dz\neq 0$$
where $z_{0}(x_{i})$ represents the neutral axis coordinates of cross-section *i*:(16)$$z_{0}(x_{i})=\frac{d-e-\omega (x_{i})}{2}$$

Owing to the variation in material removal thickness, the size of the remaining blank portion also changes with the x-coordinate. Hence, the moment of inertia $I(x_{i})$ for section *i* can be expressed as follows:(17)$$I(x_{i})=\frac{b\cdot {\left[d-e-\omega (x_{i})\right]}^{3}}{12}$$

The torque $M_{x_{i}}(e)$ causes the blank to undergo flexural deformation, resulting in a stress distribution $\sigma_{m_{i}}(z)$ within section *i* that varies linearly with thickness Z, as expressed in Equation (18).
(18)$$\sigma_{m_{i}}(z)=\frac{M_{x_{i}}(e)}{I(x_{i})}(z_{0}(x_{i})-z)$$

At this point, in section *i*, the newly remaining stress distribution that satisfies the force and moment equilibrium conditions can be represented as follows.
(19)$$\sigma_{new_{i}}(z)=\sigma_{s_{i}}(z)+\sigma_{m_{i}}(z)$$

By substituting Equations (12), (13), (15)–(18) into Equation (19), the solution yields a new stress distribution function equation within section *i*. Owing to the material removal thickness variation, the stress distribution function in each section differs. Therefore, by separately calculating the stress distribution function in each section, an n-dimensional stress function vector $\alpha^{T}$ can be obtained. This vector is used to describe the distribution in each blank section and can be expressed as follows.
(20)$$\alpha^{T}=(\sigma_{new}(z,x_{1}),\sigma_{new}(z,x_{2}),\cdots ,\sigma_{new}(z,x_{i}),\cdots ,\sigma_{new}(z,x_{n}))$$

In addition, it is possible to discretize the thickness coordinate z into m discrete values in each section and substitute these into the Equation (20). This allows us to obtain the final blank overall stress matrix $A_{m\times n}$, which is expressed as follows:(21)$$A_{m\times n}=\left(\begin{array}{ccc} \sigma_{new}(z_{1},x_{1}) & \ldots & \sigma_{new}(z_{1},x_{n})\\ \vdots & \ddots & \vdots \\ \sigma_{new}(z_{m},x_{1}) & \cdots & \sigma_{new}(z_{m},x_{n}) \end{array}\right)$$

In conclusion, when the *m* and *n* values are sufficiently small, the stress matrix $A_{m\times n}$ can be used to describe the distribution of residual stress in the x-direction at any point within the blank ZX section. The model overlooks the influence of shear stress on bending deformation. Therefore, under the uneven milling conditions, the blank bending deformation also satisfies the pure bending deformation condition. Using Equations (15) and (17), it is possible to calculate the curvature $k_{x_{i}}=M_{x_{i}}(e)/\left(E\cdot I(x_{i}\right))$ at section *i*, where *E* represents the material elastic modulus. By individually calculating the curvature equation in each section, an n-dimensional curvature vector $\beta^{T}$ can be obtained, which is expressed as follows:(22)$$\beta^{T}=(k_{x_{1}},k_{x_{2}},\cdots ,k_{x_{i}},\cdots ,k_{x_{n}})$$

In the curvature vector $\beta^{T}$, the discrete sectional curvature values cannot be directly obtained through integration to determine the blank deformation deflection curve. Therefore, it is necessary to fit the elements in curvature vector $\beta^{T}$. Specifically, it is possible to fit the curvature values $k_{x1}∼k_{xn}$ with respect to the independent variable x in the interval [0,*l*] to obtain the curvature fitting function $\rho (x),x\in \left[0,l\right]$. Subsequently, the curvature fitting function can be integrated to obtain the blank deformation deflection curve. Therefore, under the uneven milling conditions, the blank deformation deflection curve can be represented as follows:(23)$$\omega (x)=\iint \left[\rho (x\right)dx]dx+C_{3}\cdot x+C_{4}$$

Note that in Equation (23), $C_{3},C_{4}$ represents an undetermined coefficient.

## 3. Case Study Research

In recent years, researchers have increasingly focused on the shape-deviation issue that occurs during the milling process for large aluminum alloy thin-walled structural components. In this context, significant attention has been paid to the milling processes of thin-walled side walls and large frame lattice structures. The causes of such shape deviations include the workpiece thin-walled characteristics and improper clamping positions, which hinder effective machining deformation suppression caused by mechanical thermal and cutting force loads during the manufacturing process. These shape deviations are primarily attributed to the surface residual stress and thermoelastic effects of thin-walled components [27,28]. Therefore, addressing these challenges is crucial for enhancing machining quality and workpiece precision.

### 3.1. Experimental Specimens

This study proposes a model for predicting the evolution of overall residual stress and machining deformation in beam-type components under uneven thickness milling conditions. To validate the accuracy of the proposed model, 7050-T7451 aluminum alloy rolled plates were selected as experimental specimens. This material possesses the following mechanical properties: an elastic modulus of 71.7 GPa and a Poisson’s ratio of 0.33. As shown in Figure 3, a specimen of 38 mm thickness was divided into two parts (1 and 2) using wire electrical discharge machining. Part 1, with dimensions of 150 mm length, 100 mm width, and 38 mm thickness, was utilized to determine the initial distribution of residual stress in the blank overall thickness. Part 2, with dimensions of 680 mm length, 100 mm width, and 38 mm thickness, was used for mechanical machining experiments. Both specimens were assumed to have an identical initial distribution of residual stress and mechanical properties, and their longitudinal directions align with the aluminum plate rolling direction.

### 3.2. Initial Residual Stress Measurement

In this research, the contour method was employed to measure the IRS longitudinal distribution along the specimen thickness (part 1). Compared to the layer removal and the crack compliance methods, the contour method can be used to determine residual stress throughout the cross-section of complex geometries [20,31]. The main procedure is illustrated in Figure 4: (1) The Sodick AQ560L wire electrical discharge machining (EDM) machine was utilized, with a 0.15 mm diameter brass wire for slow wire cutting, at a cutting speed of 0.5 mm/min, to cut the specimen (part 1) into two sections along the central line. (2) Owing to the deformation caused by the release of internal residual stress in the cut surface, the HEXAGON Leitz PMM-C CMM (measurement accuracy is 0.3 μm) was used to measure the cut surface Ux displacement. The measurement points were spaced 5 mm apart in the width direction and 1 mm apart in the thickness direction. Subsequently, the displacement data were fitted using a fourth-order Fourier series ($U_{x}=a_{0}+\sum_{n=1}^{4}\left[a_{n}\cos\left(n\cdot w\cdot z\right)+b_{n}\sin\left(n\cdot w\cdot z\right)\right]$), where $a_{0},a_{n},b_{n},w$ are fitting parameters, and z represents the coordinate along the thickness direction. (3) The measured displacements were applied as reversed boundary conditions (−Ux) vertically to the nodes on the cut surface in the finite element model. The finite element model employed C3D8R hexahedral elements for meshing, of 1 × 1 × 1 mm size for each element. To avoid rigid body displacement, fixed constraints were applied on the opposite side of the model’s cut surface. Finally, the IRS longitudinal (rolling direction) distribution was obtained using numerical calculations, as shown in Figure 5.

### 3.3. Residual Stress Measurement at a Local Level

Current research indicates that the residual stress induced by machining is predominantly present near the material surface (within a few hundred μm) [32], and contributes minimally to the deformation of thick aluminum alloy components. This study proposes a predictive model for the evolution of overall residual stress in workpiece uneven milling (detailed in Section 2.2). To validate the model’s predicted stress accuracy, this experiment uses the blind-hole method to measure the local residual stress state before and after workpiece milling, and compares it with the theoretical model and simulation results. The blind-hole method primary procedure for stress measurement includes: (1) Initially, the test workpiece surface is polished flat using 600-grit sandpaper and cleaned with anhydrous ethanol to ensure a clean, grease-free surface. (2) ZEMIC’s BE120-2CA-K-Q30P1.5K strain gauges were affixed to the cleaned surface and calibrated. (3) MTS3000 Resten equipment and a 1.8 mm diameter drill bit from the SINT Company were used to detect local residual stress. The experimental setup is shown in Figure 6.

To characterize the local stress state before and after milling, a drilling depth of 2 mm was selected, divided into 25 incremental steps, with a drilling depth of 0.08 mm for each step. Strain data were collected using the HBM universal data acquisition system. The residual stress distribution was calculated following the ASTM-E837-20 standard [33] procedure. A total of 13 local stress data points were collected to validate the model’s stress evolution prediction accuracy under the uneven milling conditions, as described in Section 4.1.

### 3.4. Analytical and Finite Element Model Establishment

The calculation program was written using MATLAB 2012b software to obtain the stress distribution data and the prediction results of machining deformation. Add the IRS curve (as shown in Figure 4) and material parameters (Section 3.1) to the calculation program as initial parameters. To predict workpiece deformation under uneven milling conditions, it is not possible to directly solve and obtain the deflection curve using the integration method (Section 2.2). Therefore, a fitting process was implemented in the software program to apply each section’s obtained curvature. The fitting method utilizes a fourth-degree polynomial as follows $\rho (x)=a\cdot x^{4}+b\cdot x^{3}+c\cdot x^{2}+d\cdot x^{1}+e$, where a,b,c,d,e are the fitting parameters, and x represents the coordinate along the longitudinal direction.

To verify the theoretical analysis model’s accuracy, this study employed the ABAQUS general finite element analysis software package to construct a finite element model under the uneven milling conditions. As shown in Figure 7, this model is divided into three submodels, with the main steps as follows:(1)First, based on the element birth and death method, submodel one is established to satisfy the even thickness removal condition. Subsequently, utilizing the SIGINI subroutine, the IRS function shown in Figure 4 is applied to the nodal points of each element in submodel one. This facilitates the simulation calculation for the first layer even thickness removal.(2)Second, using the restart request techniques and manual remeshing techniques, the deformed entity resulting from the submodel one calculation is imported into submodel 2. Subsequently, by employing the element birth and death method, the mesh is partitioned and the model attributes are imposed. Moreover, using the Map Solution keyword, the submodel one stress result data are mapped onto the nodal points of each element in submodel two. This completes the simulation calculation for the second layer uneven removal.(3)Finally, following the Step 2 procedure, the deformed entity and stress result data from the submodel two calculation are mapped into submodel three. As a result, the simulation calculation for the third layer uneven removal was accomplished. In the aforementioned finite element model, C3D8R hexahedral elements with dimensions of 5 × 5 × 0.5 mm were utilized. Meanwhile, the following boundary conditions were defined: the displacement constraint on the left end of the model’s plane (Ux = Uy = Uz = 0), and the vertical (Z-direction) and horizontal (Y-direction) displacement on the right end (Uy = Uz = 0).

### 3.5. Experiment

This study also validates the effectiveness of the analytical model using milling experiments. Previous research has indicated that increasing the tool diameter [34], reducing the cutting depth [35], and lowering the feed rate [36] can effectively reduce the MIRS introduced during the milling process. To balance processing efficiency, the same milling parameters were utilized throughout the process. Please refer to Table 1 for specific cutting parameters. The milling experiments were carried out on a VMC850B vertical machining center, employing a BT40-400R-80-120 milling cutter with a diameter of 80 mm fitted with five APMT1604-M2 carbide inserts.

As depicted in Figure 8c, IRS measurement was performed on part 1. The specific measurement process can be found in Section 3.2. Part 2, being the material used for the milling experiments, is assumed to have the same distribution of residual stress and material characteristics as the residual stress measurement sample (part 1). The experiment necessitates the removal of three layers of material, each with a thickness of e = 6 mm, from the top surface of the workpiece (part 2). As indicated in Table 1, the cutting tool has a depth of cut of 1mm, and for each individual operation, the cutting width is 50 mm. The removal process for each layer of material entails processing of 12 consecutive times.

Prior to the start of machining, the workpiece did not undergo any initial deformation. Therefore, when removing the first layer of part 2, the cutting depth during the milling process remains constant (uniform milling). However, when removing the second and third layers, the specimen will experience warping deformation due to the coupled effects of redistributed internal residual stress and thermal-elastic effects during machining [27,28]. This will result in a variation in the local cutting depth during the material removal process (uneven milling).

Figure 8b demonstrates the workpiece clamping scheme. Before removing the second and third layers of material, the workpiece is positioned by a gauge block to ensure that the lowest point of the workpiece bottom is tangential to the workbench surface. The positioning height is determined by the maximum deflection value of the workpiece after removing the previous layer of material. Finally, the workpiece is secured to the workbench using a wedge fixture in a lateral top-down manner. After each milling operation, the deformation deflection values of the upper and lower surfaces of part 2 are measured using the HEXAGON EXPLORER CMM (with a measurement accuracy of 0.005 mm). After the milling process, residual stress within a depth range of 2 mm is measured on 13 sampling points of the workpiece using a drilling method. The specific measurement scheme is detailed in Section 3.3, and the locations of the sampling points can be found in Section 4.1.

## 4. Results and Discussion

### 4.1. Model Stress Results Comparison

Figure 9a illustrates the comparative results of stress distribution along the workpiece thickness direction, obtained from the analytical model and the finite element model, after the removal of the first layer removal (workpiece milled from a 38 to 32 mm thickness at 6 mm milling depth). The analytical model predicts an increase of 11.78 MPa in compressive stress on the workpiece upper surface (located at position D in Figure 9a, z = 32 mm) relative to the IRS, while an increase of 4.80 MPa in tensile stress occurred on the workpiece lower surface (located at position A in Figure 9a, z = 0 mm) relative to the IRS. In the finite element model, these increments are slightly lower, measuring 11.61 MPa and 4.67 MPa, respectively. This phenomenon can be explained by the workpiece stress redistribution after the first layer removal, leading to a concave deformation and resulting in compressive stress on the upper surface and tensile stress on the lower surface. It is worth noting that at position (B), located at one-fourth of the initial thickness of the workpiece (d = 38 mm) (as shown in Figure 9a, z = 9.5 mm), no significant change in residual stress was observed before and after the first layer removal. Figure 9b presents the experimental stress data for points A, B, C, and D after the first layer removal. The analytical model demonstrates an average relative error of 34.28% compared to the experimental data, while the average absolute error is only 1.23 MPa. This discrepancy between experimental and analytical predictions is attributed to the combined effects of stress measurement system errors and residual stress induced by the milling process. It is evident that both the analytical and finite element models yield similar stress data, and the analytical model predictions exhibit good agreement with the experimentally measured stress data, further confirming the analytical model’s accuracy in predicting stress evolution and workpiece deflection during the layer milling process.

During the aircraft beam-like component machining process, the significant dimensions and thin-wall characteristics of parts hinder the effective suppression of deformation through process connections. As a result, variations in cutting depth are quite common throughout machining. Upon the first layer removal, the redistribution of overall residual stress and the introduction of additional residual stress in the machining process disrupt the internal stress equilibrium of the parts, leading to warping deformations and subsequent fluctuations in local cutting depths during the milling process. Figure 10a,c display the surface plots of the overall residual stress distribution in the z-axis (thickness direction) and x-axis (length direction) sections, respectively, of the analytical and finite element models after the second and third layer removal. To eliminate uncertainties in edge stresses caused by boundary conditions in the finite element model, the stress surface values in the simulation model range from 30 to 650 in the length direction. After the second layer removal, the analytical model predicts a maximum residual stress of 13.202 MPa in the part and a minimum residual stress of −8.974 MPa. Following the third layer removal, the maximum residual stress in the part is 12.123 MPa, while the minimum residual stress is −10.013 MPa.

The analytical model predicts a slight discrepancy in the overall residual stress distribution on the ZX section of the workpiece compared to the stress results of the finite element model after uneven milling. Figure 10b,d illustrate the stress error plots (i.e., difference between the analytical and finite element models) after the second and third layer removals, respectively. It can be observed that overall, the absolute error in stress prediction between the analytical and finite element models does not exceed 1 MPa. However, in Figure 10b, the largest difference occurs between the bottom and top positions, with a gap of 1.61 MPa. In Figure 10d, the maximum difference appears only at the bottom position, with a gap of 1.90 MPa. Compared to the finite element model, the average error in stress distribution prediction on the ZX section of the workpiece after uneven milling is 3.40%, with a maximum error of 4.94%.

To verify the stress distribution accuracy predicted by the analytical model, Figure 11 shows a comparison between the analytical model, finite element model, and the experimental results after the third layer milling completion. Figure 11a–c present the comparative stress results of the nine measurement points. Figure 11d shows the coordinates of the nine measurement points. The analytical model predicts an uneven stress distribution at the workpiece middle position (x = 340 mm) and the two workpiece end positions (x = 30 mm, x = 650 mm). Specifically, at point A2, the stress is 1.59 MPa lower compared to points A1 and A3, while at point B2, the stress is 1.9 MPa higher compared to points B1 and B3. Similarly, at point C2, the stress is 0.73 MPa lower compared to points C1 and C3. When compared to the experimentally measured stress data, certain discrepancies emerge in the analytical model predictions, with average differences of 2.27, 1.76, and 2.57 MPa, respectively. This is primarily caused by measurement errors in the stress detection process and the incomplete release of stress after milling. However, it is evident that the analytical model exhibits a high level of consistency with the experimental measurements in terms of stress distribution prediction. In summary, under uneven milling conditions, there exists a difference of 1 MPa to 3 MPa in the stresses between the middle and end positions of the workpiece. This difference can be attributed to the fact that during the uneven removal process, the curvature radius at the middle position of the workpiece experiences the least variation (greater degree of bending), while the curvature radius at the ends of the workpiece gradually increases (lesser degree of bending). As a result, the workpiece experiences an uneven distribution of bending stresses.

### 4.2. Deformity Prognostication Outcome Comparison

After each layer of milling is completed, the deformation deflections of three different points on both the upper and lower surfaces of the workpiece are measured using a coordinate measuring machine (CMM). The average value of these measurements is taken as the experimental result. Figure 12 illustrates a comparative analysis of the analytical model, finite element model, and experimental results regarding deflection. In Figure 12a, the deformity comparison on the workpiece surface is shown representing the workpiece deflection values after each layer is removed. Meanwhile, Figure 12b demonstrates the maximum deflection comparison on the workpiece bottom surface, illustrating the maximum deflection values obtained through the accumulation of deformations after each layer is removed. Owing to the blank symmetry, both the analytical and finite element models exhibit warping at the workpiece ends, with the maximum deflection occurring in the middle region. To minimize experimental errors, the average of two sets of measured maximum deflection values is considered for both the upper and lower surfaces of the workpiece. Additionally, Figure 12 presents the deformation results of the analytical and finite element models under the uniform layer removal condition, enabling a comparison between the uniform and uneven milling scenarios.

According to Figure 12, although the measured deformity is smaller than the deformity predicted by the analytical model, its trend corresponds to the analytical model results. The comparison on the upper-surface deformity results reveals that the deformity values predicted by the analytical model for the three milling layers are 0.416, 0.531, and 0.205 mm, whereas the experimentally measured deformity values are 0.362, 0.483, and 0.168 mm, respectively. As for the lower surface deformity results, the analytical model predicts deformity values of −0.416, −0.947, and −1.152 mm, while the experimental measurements yield deformity values of −0.332, −0.695, and −0.791 mm, respectively. Compared to the experimental results, the analytical model relative errors for the maximum deformity on the upper and lower surfaces are 19.8% and 45.6%, respectively. In comparison to the finite element model, the average error is 0.84%. It is important to note that both the analytical and finite element models do not consider the influence of clamping forces, cutting thermal loads, or residual stress induced by machining, which are the primary sources of deviation between the experimental and analytical model results. In practical machining processes, these factors, cannot be completely eliminated in terms of their impact on machining deformation.

Furthermore, noticeable differences in the deformity are also observed on the workpiece upper and lower surfaces under both the uniform and uneven milling scenarios. The comparison presented in the Figure 12 illustrates the analytical and finite element models assessment of the maximum deflection in these two cases. In the analytical model, the deflection results for uneven milling exhibit a discrepancy in comparison to the uniform layer removal, with a maximum deformation error of 26.5% and a maximum difference of 0.094 mm on both workpiece surfaces. It is foreseeable that this difference will further increase as the material is progressively removed layer by layer. In comparison to the finite element model, the analytical model yields an average error of 0.58% for predicting workpiece deflection under the aforementioned scenarios.

It should be noted that the workpiece deformity is closely correlated with the stress state of the material being removed. When the first material layer is removed (milling thickness interval z ∈ [32, 38]), according to the results shown in Figure 4, the removed portion experiences a compressive stress state. This leads to warping deformation development in the workpiece, resulting in an increase in compressive stress at the workpiece top position to 25.67 MPa, as depicted in Figure 9a. Consequently, when the second material layer is removed (milling thickness interval z ∈ [26, 32]), the workpiece deformity further increases. However, as shown in Figure 10a, when the third material layer is removed (milling thickness interval z ∈ [20, 26]), the removed portion remains in a compressive stress state, albeit with compressive stresses not exceeding 8 MPa. As a result, the workpiece deformity significantly reduces. Thus, it can be concluded that the variation in workpiece deformity, increasing and subsequently decreasing, is closely related to the changing stress state of the material being removed. The greater the stress amplitude, the more significant the workpiece deformation.

In recent years, in-depth investigations have been conducted into the correlation between workpiece layout and deformation in raw materials. The research contends that the deformation caused by processing at the raw material center is generally smaller compared to the edges [19,20]. Moreover, researchers have proposed methods to correct the machining datum prior to precision machining to control deformation during subsequent processing. Research demonstrates that this approach effectively reduces final component deformation [17,18]. However, during the process of step-by-step milling from top to bottom, the deflection of the workpiece upper and lower surfaces varies. With each layer removed, the deformation of the raw material upper surface will be eliminated, resulting in an imbalance of internal forces and subsequent distortion of the workpiece. This distortion can be described by the maximum deflection value of the material’s upper surface after each layer is removed (see Figure 12a). Nevertheless, the lower surface has not undergone the milling and removal processes, thus, its deformation is not eliminated. Namely, the deformation after each layer removal accumulates, and the incremental deflection of the lower surface is equal to the maximum deflection value of the upper surface after each layer is removed (i.e., the summation of the maximum deflection values of the upper surface). This deflection value exhibits a linear upward trend. Therefore, it can be argued that the cumulative effect of deformation on the lower surface exacerbates the final workpiece deformation, and by eliminating this deformation, processing-induced distortion can be controlled. This perspective is consistent with previous research [17,18].

In conclusion, the comparative results have validated the accuracy of the analytical model. This model precisely predicts the final deformation outcomes of the workpiece, aligning well with finite element simulation and experimental measurement results. Moreover, the analytical model quantitatively characterizes the residual stress distribution map within the cross-sections of the workpiece during the layer-by-layer removal process and exhibits a remarkable consistency with the finite element simulation and experimental measurement results. It accurately predicts the evolution of residual stresses and machining deformation in beam-like aluminum alloy components caused by variations in cutting depth during processing.

It should be noted that the proposed model in this study has certain limitations. It is based on the assumptions mentioned in Section 2 and solely considers the uniaxial residual stress evolution of beam-like aluminum alloy components under uneven milling conditions. The evolution pattern of biaxial residual stresses under uneven milling conditions has not been taken into account, and this aspect remains a key area for further improvement of the model.

## 5. Conclusions

In this study, A concept of uneven milling during the machining process of aluminum alloy components has been innovatively introduced. This concept refers to the unevenness in material removal thickness caused by variations in local cutting depth of the workpiece. Furthermore, an analytical model has been developed to predict the overall evolution of residual stresses and machining deformations in beam-like aluminum alloy components under uneven milling conditions. Finite element simulations and practical machining experiments were conducted to validate the accuracy of the proposed analytical model. By comparing the calculated results from the analytical model with experimental data, the following conclusions can be summarized:The analytical model proposed in this study demonstrates accurate prediction of machining deflection and residual stress distribution on the ZX cross-section of beam-like aluminum alloy components under uneven milling conditions. When compared to finite element simulation results, the analytical model exhibits an average prediction error of 3.40% for residual stress distribution within the ZX cross-section and an average prediction error of 0.71% for machining deflection. Comparison with experimental measurements further confirms the good consistency between the predictive results of the analytical model and the experimental observations.Under the conditions of uneven milling, there exists a difference of 1 MPa to 3 MPa in residual stress between the middle and end positions of the aluminum alloy component. This difference is attributed to the varying curvature radii along the component during the uneven material removal process. The middle position of the component has the smallest curvature radius (higher degree of bending), while the curvature radii gradually increase towards the ends (lower degree of bending). As a result, the distribution of bending stress is uneven across the component. However, the existing literature on layer milling theory has not provided a sufficient description of this difference.During the layer-by-layer milling process, the magnitude of the workpiece deformation is influenced by the stress state of the material being removed. The greater the stress amplitude, the more significant the workpiece deformation. Under uneven milling conditions, there is a difference in deformation deflection between the upper and lower surfaces of the workpiece. The deformation on the upper surface is removed in the subsequent milling steps, while the lower surface undergoes successive layers of milling without the removal of previously deformed material. This accumulation of deformation deflection on the lower surface leads to further exacerbation of the overall machining deformation of the workpiece.

It should be noted that the proposed model in this study has certain limitations. It is based on the assumptions mentioned in Section 2 and solely considers the uniaxial residual stress evolution of beam-like aluminum alloy components under uneven milling conditions. The evolution pattern of biaxial residual stresses under uneven milling conditions has not been taken into account, and this aspect remains a key area for further improvement of the model.

## Figures and Tables

**Figure 1 materials-16-06113-f001:**
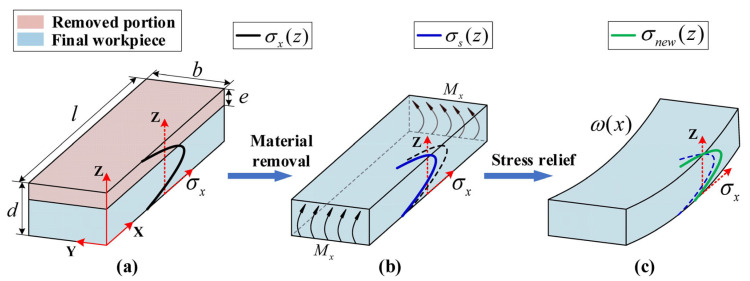
Machining deformation mechanism induced by uniform milling. (**a**) initial state of the workpiece. (**b**) torque generated after material removal. (**c**) deformation occurs after stress release.

**Figure 2 materials-16-06113-f002:**
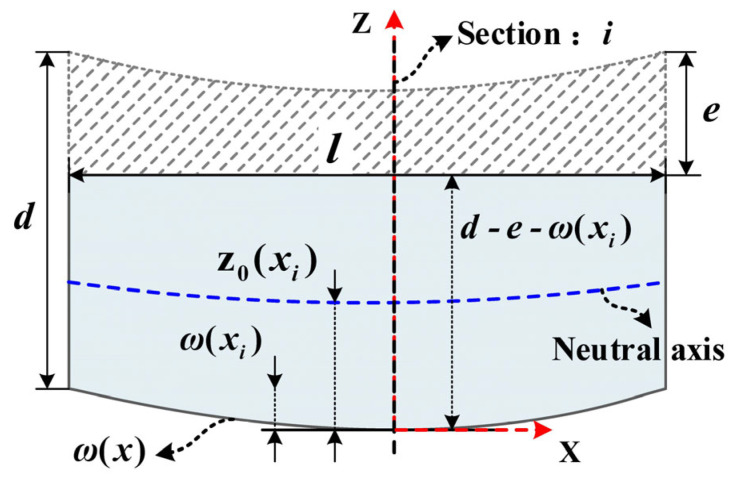
Workpiece uneven milling mechanism from a Y-oriented perspective.

**Figure 3 materials-16-06113-f003:**
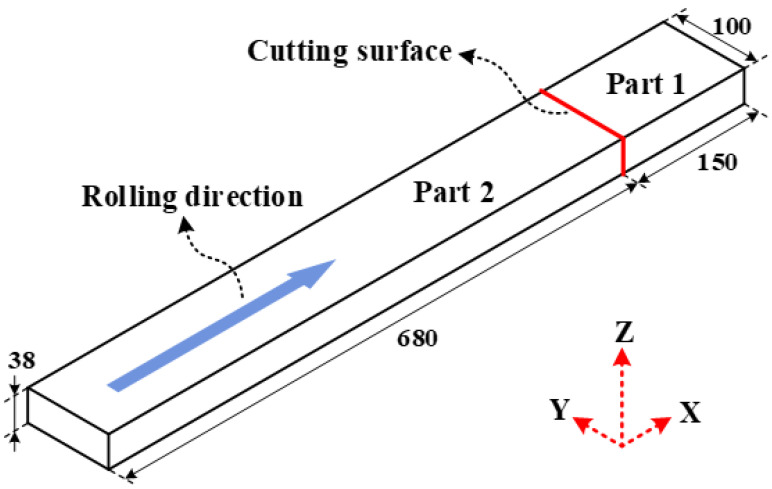
Experimental specimens.

**Figure 4 materials-16-06113-f004:**
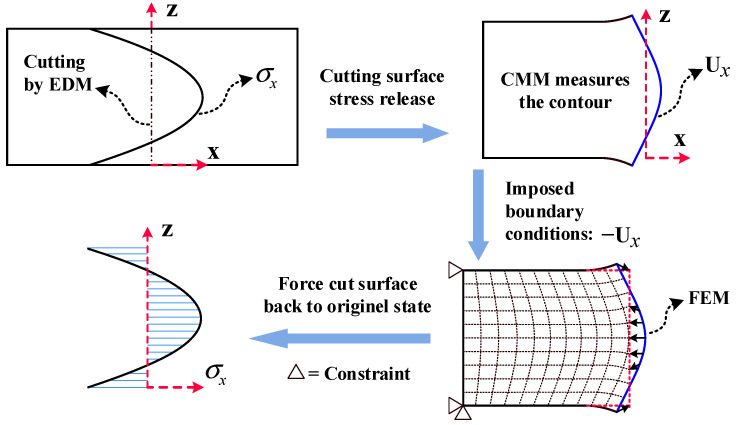
Stress evaluation method using contour analysis.

**Figure 5 materials-16-06113-f005:**
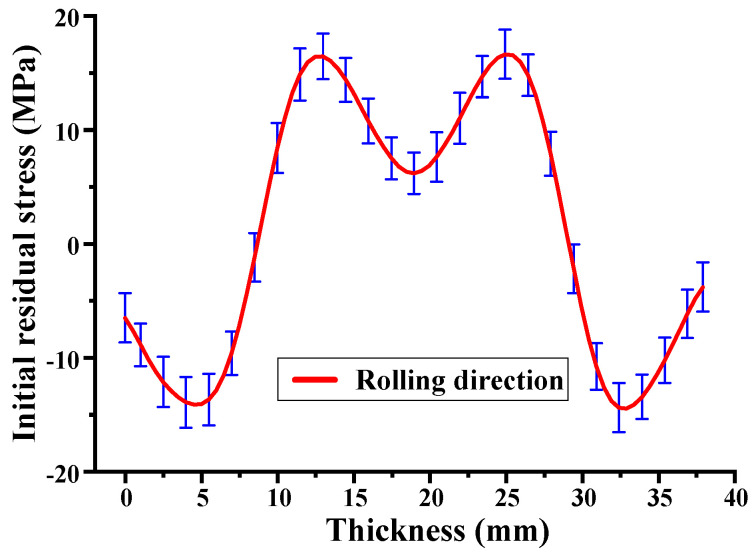
Initial residual stress curve.

**Figure 6 materials-16-06113-f006:**
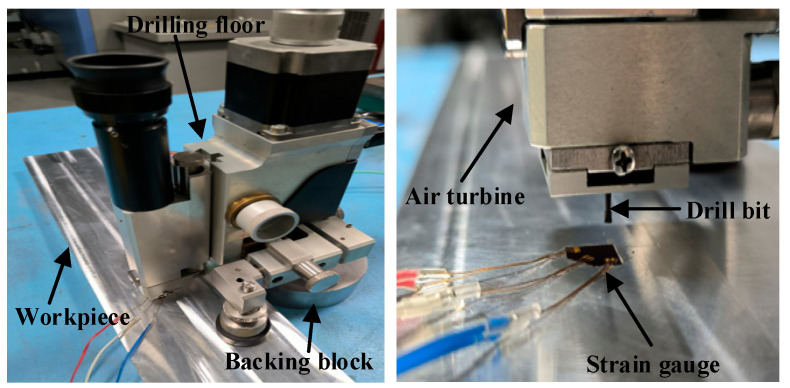
Stress assessment equipment for utilizing blind-hole methodology.

**Figure 7 materials-16-06113-f007:**
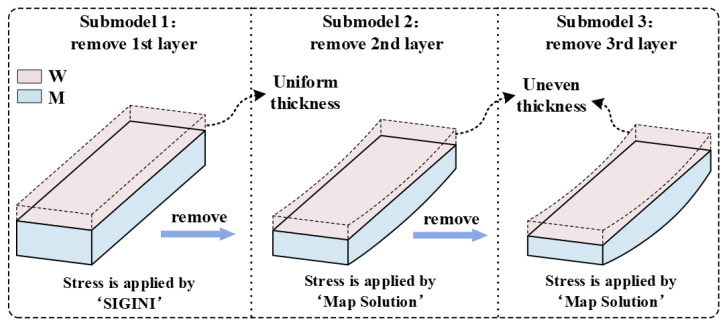
Finite element model establishment.

**Figure 8 materials-16-06113-f008:**
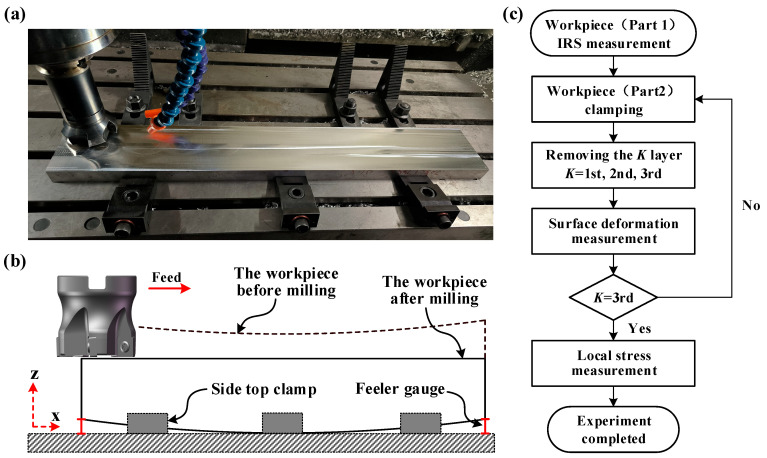
Machining experiment. (**a**) manufacturing process. (**b**) Clamping scheme. (**c**) process flowchart.

**Figure 9 materials-16-06113-f009:**
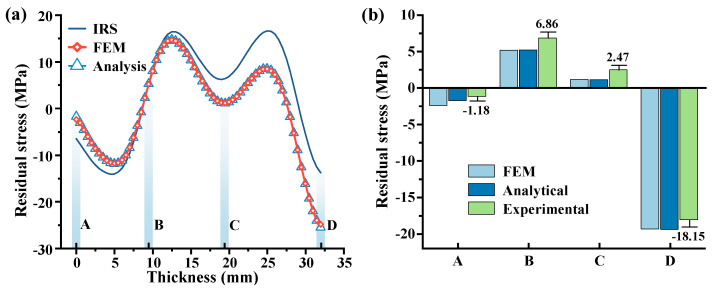
Comparison between the finite element model, analytical model, and experimental stress data after 1st layer removal. (**a**) Comparison of stress results between analytical model and finite element model. (**b**) Comparison of stress results between analytical model, finite element model, and experimental measurements at points A, B, C, and D.

**Figure 10 materials-16-06113-f010:**
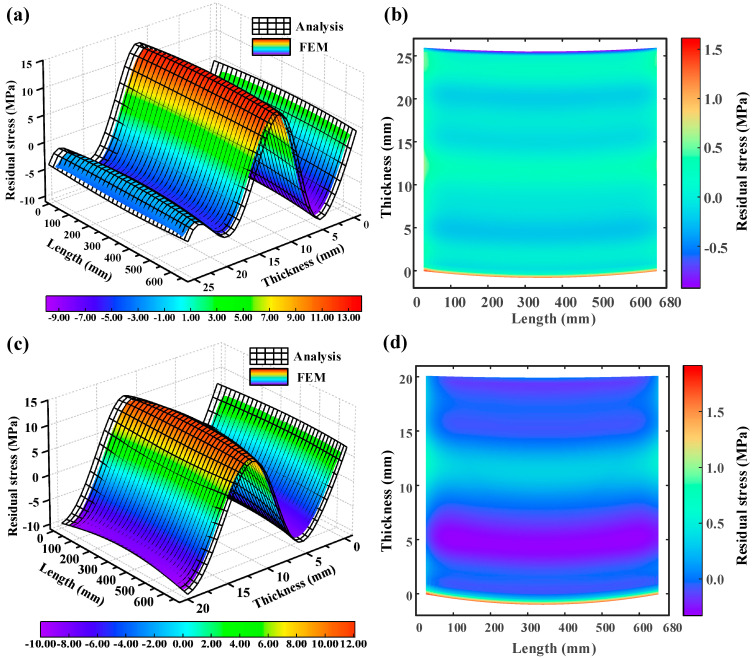
Comparison of residual stress distribution between the analytical model and the finite element model. (**a**) Comparison of residual stress distribution after removing the second layer. (**b**) Stress error distribution between the analytical model and the finite element model after removing the second layer. (**c**) Comparison of residual stress distribution after removing the third layer. (**d**) Stress error distribution between the analytical model and the finite element model after removing the third layer.

**Figure 11 materials-16-06113-f011:**
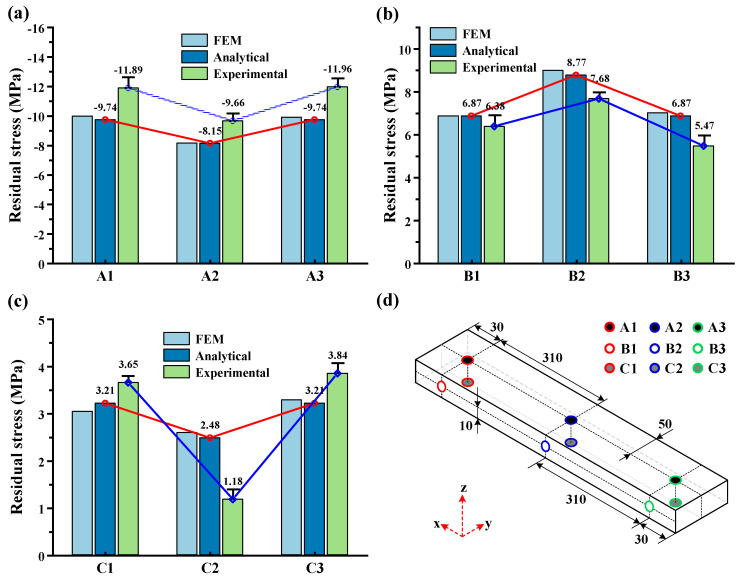
Comparison of local residual stress between analytical model, finite element model, and experimental results. (**a**) Comparison of local residual stresses at points A1, A2, and A3. (**b**) Comparison of local residual stresses at points B1, B2, and B3. (**c**) Comparison of local residual stresses at points C1, C2, and C3. (**d**) Distribution of Measurement Points.

**Figure 12 materials-16-06113-f012:**
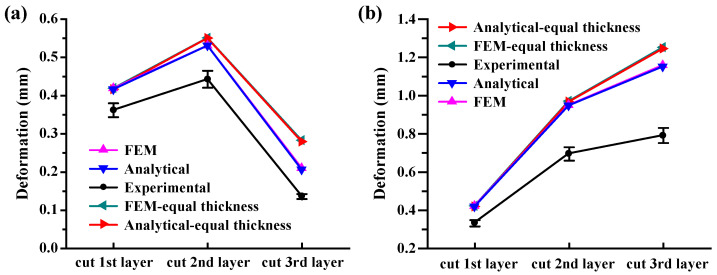
Final deformation results comparison among the analytical model, finite element model, and experimental data. (**a**) Comparison of the deformation results on the top surface of the artifact. (**b**) Comparison of the deformation results on the bottom surface of the artifact.

**Table 1 materials-16-06113-t001:** Cutting parameters.

Diameter (mm)	*Z*	*V*_c_ (m/min)	*f*_z_ (mm/tooth)	*V_f_* (mm/min)	*a*_p_ (mm)	*N* (r/min)
80	5	300	0.1	597	1	1193

*Z* number of teeth, *V*_c_ cutting speed, *f*_z_ feed rate per tooth, *V_f_* feed rate, *a*_p_ depth of cut, *N* spindle speed.

## Data Availability

The data presented in this study are available in the article.

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
