# Peer review of "Prediction Model for the Evolution of Residual Stresses and Machining Deformation of Uneven Milling Plate Blanks"

_materials, 2023, doi:10.3390/ma16186113_

Round 1

Reviewer 1 Report

The paper introduces an analytical model aimed at predicting the evolution of residual stress and machining deformation during uneven milling conditions. The provided results and accompanying discussion are thoroughly discussed, effectively presenting the rationale for modelling stress states and residual stresses within the report. The paper's overall presentation and structure are commendable, supported by well-justified result explanations that engage readers effectively. Although proofreading is suggested but the work contributes to the existing knowledge in machining science, making the work satisfactory in its current state.

The strength was how well the numerical model has been presented and implemented in the work.

The weakness was the lack of a detailed design of experiments and the statistical aspect of the experimental work done.

Reviewer 2 Report

This quality paper has been worked on thoroughly, particularly regarding its technical details. However, there are a few points that need attention before it can be considered for publication:

1.       The strengths and limitations of the applied approach should be clearly identified for the readers of the paper.

2. Some of the bullet points in the conclusion are simplistic; Please try to emphasize your novelty, put some quantifications, and comment on the limitations.  

   3. The introduction (background section) may be enlarged as well, with a couple of related studies.

4. The selection of machining parameters and their levels are on what basis?

5. No repetition of experiments has been carried out. Authors should conduct experiments thrice and show the results using error bars.

6. Abstract is reflect the content and summarizes the problem, the method, the results, and the conclusions.

7.      Improving the writing and typo errors to make it qualified and readable. Please carefully check the sentences again.

Moderate editing of English language required.

Reviewer 3 Report

The study contains innovative results in general terms. It is possible to say that the study has been prepared with sufficient care. However, some of the assumptions taken into account in the construction of the mathematical model (shown on page 4) should be supported by more literature. Also, some revision suggestions are shown in the text. Please refer to the pdf file for this.
